# Can Decomposition Approaches Always Enhance Soft Computing Models? Predicting the Dissolved Oxygen Concentration in the St. Johns River, Florida

**Mohammad Zounemat-Kermani** [1] **, Youngmin Seo** [2] **, Sungwon Kim** [3,*] **, Mohammad Ali Ghorbani** [4] **, Saeed Samadianfard** [4] **, Shabnam Naghshara** [4] **, Nam Won Kim** [5] **and Vijay P. Singh** [6]

[1] Department of Water Engineering, Shahid Bahonar University of Kerman, Kerman 76169-14111, Iran; zounemat@uk.ac.ir

[2] Department of Constructional and Environmental Engineering, Kyungpook National University, Sangju 37224, Korea; ymseo@knu.ac.kr

[3] Department of Railroad Construction and Safety Engineering, Dongyang University, Yeongju 36040, Korea

[4] Department of Water Engineering, University of Tabriz, Tabriz 51666-16471, Iran; ghorbani@tabrizu.ac.ir (M.A.G.); s.samadian@tabrizu.ac.ir (S.S.); shabnam.naghshara@gmail.com (S.N.)

[5] Department of Land, Water and Environment Research, Korea Institute of Civil Engineering and Building Technology, Goyang 10223, Korea; nwkim@kict.re.kr

[6] Department of Biological and Agricultural Engineering & Zachry Department of Civil Engineering, Texas A & M University, College Station, TX 77843-2117, USA; vsingh@tamu.edu

* Correspondence: swkim1968@dyu.ac.kr; Tel.: +82-54-630-1241

**Abstract:** This study evaluates standalone and hybrid soft computing models for predicting dissolved oxygen (DO) concentration by utilizing different water quality parameters. In the first stage, two standalone soft computing models, including multilayer perceptron (MLP) neural network and cascade correlation neural network (CCNN), were proposed for estimating the DO concentration in the St. Johns River, Florida, USA. The DO concentration and water quality parameters (e.g., chloride (Cl), nitrogen oxides (NOx), total dissolved solid (TDS), potential of hydrogen (pH), and water temperature (WT)) were used for developing the standalone models by defining six combinations of input parameters. Results were evaluated using five performance criteria metrics. Overall results revealed that the CCNN model with input combination III (CCNN-III) provided the most accurate predictions of DO concentration values (root mean square error (RMSE) = 1.261 mg/L, Nash-Sutcliffe coefficient (NSE) = 0.736, Willmott's index of agreement (WI) = 0.919, $R^2$ = 0.801, and mean absolute error (MAE) = 0.989 mg/L) for the standalone model category. In the second stage, two decomposition approaches, including discrete wavelet transform (DWT) and variational mode decomposition (VMD), were employed to improve the accuracy of DO concentration using the MLP and CCNN models with input combination III (e.g., DWT-MLP-III, DWT-CCNN-III, VMD-MLP-III, and VMD-CCNN-III). From the results, the DWT-MLP-III and VMD-MLP-III models provided better accuracy than the standalone models (e.g., MLP-III and CCNN-III). Comparison of the best hybrid soft computing models showed that the VMD-MLP-III model with 4 intrinsic mode functions (IMFs) and 10 quadratic penalty factor (VMD-MLP-III (K = 4 and $\alpha$ = 10)) model yielded slightly better performance than the DWT-MLP-III with Daubechies-6 (D6) and Symmlet-6 (S6) (DWT-MLP-III (D6 and S6)) models. Unfortunately, the DWT-CCNN-III and VMD-CCNN-III models did not improve the performance of the CCNN-III model. It was found that the CCNN-III model cannot be used to apply the hybrid soft computing modeling for prediction of the DO concentration. Graphical comparisons (e.g., Taylor diagram and violin plot) were also utilized to examine the similarity between the observed and predicted DO concentration values. The DWT-MLP-III and VMD-MLP-III models can be an alternative tool for accurate prediction of the DO concentration values.

**Keywords:** multilayer perceptron; cascade correlation; dissolved oxygen concentration; water quality prediction; discrete wavelet transform; variational mode decomposition

## 1. Introduction

Water quality explains the descriptions of biological, chemical, and physical characteristics of water bodies [1,2]. The assessment of water quality parameters, such as dissolved oxygen (DO), algae, nitrogen (N), total nitrogen (TN), phosphorus (P), total phosphorus (TP), biochemical oxygen demand (BOD), and chemical oxygen demand (COD), is necessary to improve the operational performance and develop water resource management effectively [3].

DO, as one of the water quality parameters, refers to the free-level non-compound oxygen dissolved in water or other liquids [4–8]. Too high or low levels of DO concentration can affect the maintenance of water quality [9]. Therefore, DO is a key parameter to assess water quality in rivers, reservoirs, and lakes, and as one of the indicators for the healthy functioning of aquatic ecosystems [10].

Water quality modeling includes contaminant transport, biochemical transformation, and forecasting/prediction of water pollution [3,11]. Accurate forecasting/prediction of water quality can provide the basic data to control water quality and deal with water quality incidents [12]. In traditional water quality modeling, water quality models can be calibrated using the trial and error method. However, since the traditional calibration processes require a large number of iterations, they cannot be effective [11]. Therefore, the inverse methods (e.g., fuzzy logic, Bayesian inference, and maximum entropy technique, etc.), which are more robust, objective, and sound approaches, are recommended. One of the widely used inverse methods is the indirect inverse method, which is developed using forging nonlinear optimization problems [11,13].

The diverse researches on water quality modeling using heuristic approaches (e.g., artificial neural network (ANN), adaptive neuro-fuzzy inference systems (ANFIS), and support vector machines (SVM), etc.) have been investigated and reported in the literature during the past two decades [14–20].

Diamantopoulou et al. [21] suggested the cascade correlation neural network (CCNN) to predict the missing monthly data of water quality parameters. Kuo et al. [22] suggested the backpropagation neural network (BPNN) to quantify the cause-and-effect relationship in reservoir eutrophication. Zhao et al. [23] developed the BPNN to forecast water quality in reservoirs. Zou et al. [11] developed the genetic algorithm-based neural network (GA-NN) to solve inverse water quality problems. Dogan et al. [24] predicted the BOD concentration using the feedforward neural network (FFNN) in the Melen River, Turkey. Najah et al. [3] used ANNs to predict three water quality parameters (i.e., total dissolved solids, electrical conductivity, and turbidity) in the Johor River, Malaysia.

Singh et al. [25] employed SVM (i.e., support vector classification (SVC) and regression (SVR)) to optimize a monitoring program using water quality data. Han et al. [26] used a radial basis function neural network (RBFNN) for water quality prediction in the wastewater treatment process. Gazzaz et al. [9] proposed the FFNN to predict water quality indexes in the Kinta River, Malaysia. Xu and Lie [12] provided a hybrid approach combining the BPNN and wavelet transform (WT) to develop a water quality model. Ay and Kisi [27] developed the k-means clustering-based multilayer perceptron (k-means MLP) for modeling the COD concentration in Adapazari, Turkey. Li et al. [28] implemented a hybrid model based on integrated SVR with the firefly algorithm (FFA) for prediction of the water quality indicator (WQI) over a period of 10 years in the Euphrates River, Iraq.

Among the various heuristic techniques, ANNs (e.g., BPNN, MLP, and FFNN) have been accomplished for prediction, forecasting, modelling, and estimation of DO [29,30]; BOD and DO [31]; TN, TP, and DO [32] using different input data in the river stream. Some heuristic approaches (e.g., MLP, FFNN, ANFIS, and RBFNN) have been applied to ponds, lakes, and reservoirs [5,33–37].

Singh et al. [4] introduced two ANNs for estimating water quality parameters (i.e., DO and BOD) in the Gomti River, India. Faruk [38] proposed an autoregressive integrated moving average (ARIMA)

based neural network (ARIMA-NN) to predict water quality parameters (e.g., temperature, boron, and DO) in the Büyük Menderes River, Turkey. Ay and Kisi [10] proposed the MLP, RBFNN, and multilinear regression (MLR) to estimate the DO concentration in Fountain Creek, Colorado. Han et al. [39] developed a self-organizing RBF (SORBF) to predict the DO concentration in activated sludge wastewater treatment processes. Martí et al. [40] proposed the MLP, gene expression programming (GEP), and MLR to estimate the DO concentration at a sand filter outlet. Antanasijević et al. [41] used the generalized regression neural network (GRNN) for forecasting the DO concentration in the Danube River, Serbia. Heddam [42] proposed the grid partition-based ANFIS (ANFIS-GRID), subtractive clustering based ANFIS (ANFIS-SUB), and MLR for estimating the DO concentration in the Klamath River, Oregon. Najah et al. [43] applied the ANFIS for the prediction of DO concentration. Nemati et al. [44] investigated ANFIS, MLP, and MLR to estimate the DO concentration in the Tai Po River, Hong Kong. Keshtegar and Heddam [45] developed the modified response surface method (MRSM) and MLP to estimate the DO concentration.

Although there have been many studies to estimate the DO concentration using heuristic approaches, the applications of CCNN have been limited. Olyaie et al. [8] used the MLP, RBFNN, SVM, and linear genetic programming (LGP) for predicting the DO concentration in the Delaware River, New Jersey. Tomić et al. [46] determined the extrapolation of an ANN model, which was established for predicting the DO concentration in the Danube River. In addition, a specific investigation using heuristic and decomposition approaches, for example, the hybrid approaches using DWT [47–50] and VMD [51], cannot be found with ease for DO forecasting/prediction category [12,52].

Liu et al. [48] proposed a hybrid model based on wavelet analysis (WA), least squares support vector regression (LSSVR), and an optimal improved Cauchy particle swarm optimization (CPSO) algorithm for predicting the DO concentration. The WA-LSSVR-CPSO model demonstrated a powerful and reliable approach for predicting the DO concentration in intensive aquaculture. Ravansalar et al. [50] evaluated 30-min DO concentration using the ANN and wavelet-based ANN (WANN) models in the River Calder, England. Results showed that the WANN model provided a 30-min DO concentration prediction that was comparable to ANN. Fijani et al. [51] discussed two hybrid models based on the complete ensemble empirical mode decomposition algorithm with adaptive noise (CEEMDAN), VMD, extreme learning machines (ELMs), and least squares support vector machines (LSSVMs) models for predicting chlorophyll-a (Chl-a) and DO concentrations, respectively. They found that the CEEMDAN-VMD-ELM model produced the best results to predict water quality parameters (e.g., Chl-a and DO concentration).

This paper proposes two heuristic models (e.g., MLP and CCNN) and decomposition approaches (e.g., DWT and VMD) for predicting the DO concentration in the St. Johns River, Florida, USA. The model performances are evaluated using model efficiency indexes and diagnostic analysis using graphical comparisons (e.g., Taylor diagram and violin plot). The paper is organized as follows: The second part provides methodologies, including MLP, CCNN, DWT, and VMD, respectively. The third part proposes a study area and data, and the fourth part presents results and discussion. Conclusions are found at the end of the paper.

## 2. Materials and Methods

### 2.1. Multilayer Perceptron (MLP) Neural Network Model

ANNs have been adopted as popular and well-established models in heuristic approaches. An ANN is a parallel information processing system with a set of neurons arranged in one or more hidden layers [53]. The MLP, which is an explicit form of the ANNs model, consists of three (or more) layers with an input layer (where the data are fed into the model), one (or more) hidden layer (where the data are processed to construct a model), and an output layer (where the results are generated) [54–57]. The neurons are connected by appropriate weights in each layer to the neurons in continuous layers. In this study, the sigmoid and linear activation functions, which have been commonly utilized for

modeling purposes [58–61], were utilized in the hidden and output layers, respectively. Moreover, the Levenberg–Marquardt backpropagation (LMBP) algorithm was employed for training the MLP model. The LMBP algorithm has a more accurate curve fitting ability that is applied to the input–output data [62,63]. Figure 1a shows the structure of the MLP model, where, *i*, *j*, *k* = the input layer, the hidden layer, and the output layer, respectively; $W_{kj}$ = the connection weights between the hidden and the output layers; $W_{ji}$ = the connection weights between the input and the hidden layers; $B_1$ = the bias in the hidden layer; and $B_2$ = the bias in the output layer. In addition, DO = dissolved oxygen (mg/L); Cl = chloride (mg/L); NOx = nitrogen oxides (mg/L); TDS = total dissolved solid; pH = potential of hydrogen; and WT = water temperature (°C).

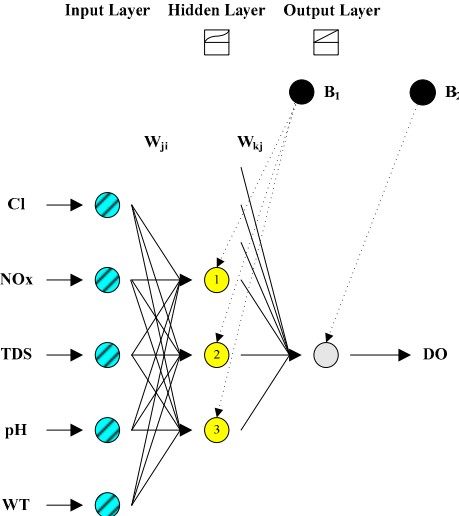

(**a**) Multilayer perceptron (MLP) neural network model (5-3-1 structure).

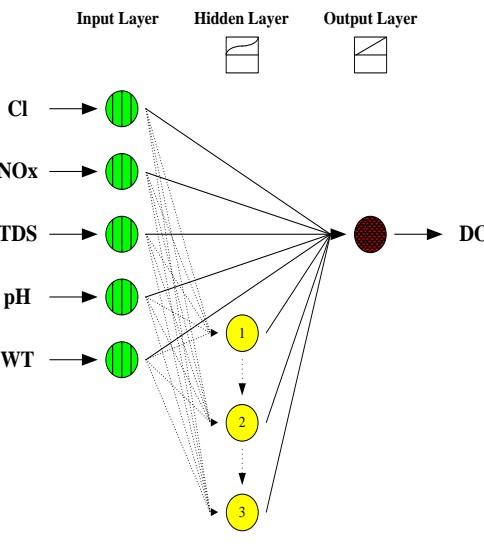

(**b**) Cascade correlation neural network (CCNN) model (5-3-1 structure)

**Figure 1.** Structure of standalone soft computing models.

### 2.2. Cascade Correlation Neural Network (CCNN) Model

A CCNN is an efficient constructive neural network combining the idea of incremental structure and learning during its training. Training starts with a minimal network consisting of an input and output layer without a hidden layer. If the training can no longer reduce the residual error, then the training phase is stopped, and enters the next phase for the training of a potential hidden neuron [64,65].

The potential hidden neuron has associated connection weights from the input layer and all preexisting hidden neurons but not toward the output layer. The connection weights are optimized by the gradient ascent method to maximize the correlation between its output and the residual error of the CCNN model. When a potential hidden neuron is trained, connection weights associated with the output layer remain unchanged. When a potential hidden neuron is added to the structure of the CCNN model, it becomes a new hidden neuron, and its incoming connection weights are fixed for the remainder of the training phase [65–67]. Figure 1b represents the structure of the CCNN model.

### 2.3. Discrete Wavelet Transform (DWT)

Wavelet transform decomposition (WTD) can be generally classified as continuous wavelet transform (CWT) and discrete wavelet transform (DWT) [68,69]. DWT requires less time of the arithmetic processes, and is easier to implement than CWT [68,70]. A fast DWT algorithm requires four filters for perfect implementation (e.g., decomposition low-pass, decomposition high-pass, reconstruction low-pass, and reconstruction high-pass) [68,71–73]. The low-pass filter for decomposition and reconstruction categories permits the interpretation of low frequency components, while the high-pass filter approves the investigation of high frequency components [72,74]. The multi-resolution approach using Mallat's DWT algorithm can be explained as a process to depict 'approximation' and 'details' for an underlying signal. An approximation produces a conventional trend of the original signal, while the details provide its high-frequency components [72,73,75]. The feature reports for Mallat's DWT algorithm can be found in Nason [76] and Percival and Walden [77]. Figure 2 shows Mallat's DWT algorithm for three-level decomposition [73].

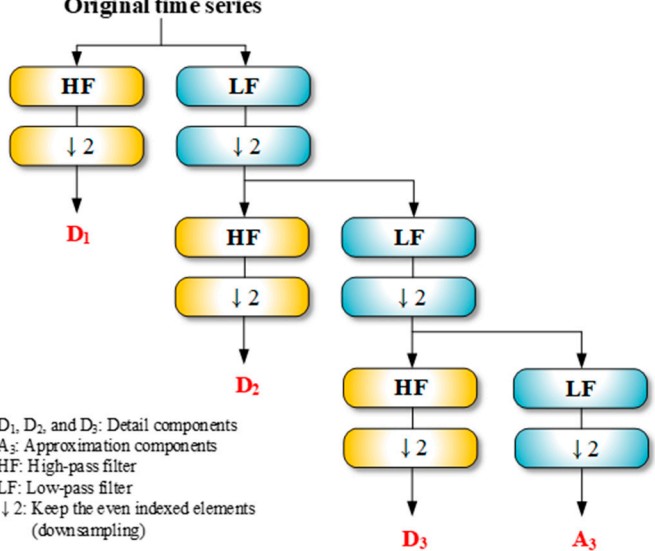

**Figure 2.** Mallat's discrete wavelet transform (DWT) algorithm for three-level decomposition [73].

### 2.4. Variational Mode Decomposition (VMD)

VMD is a fully adaptive and non-recursive algorithm for time-frequency signal analysis [78]. An original time series, $f$, can be decomposed into $K$ intrinsic mode functions (IMFs) using the VMD approach. The constrained variation formulation for generating IMFs can be written as Equation (1):

$$\min_{\{u_k\},\{\omega_k\}} \left\{ \sum_{k=1}^{K} \|\partial t \left[ \left( \delta(t) + \frac{j}{\pi t} \right) * u_k(t) \right] e^{-j\omega_k t} \|_2^2 \right\}, \text{ s.t. } \sum_{k=1}^{K} u_k(t) = f, \tag{1}$$

where $\delta$ = the Dirac function; $j^2 = -1$; $\|\cdot\|_2$ = the $L_2$ distance; $\omega_k$ = the center frequency; * = the convolution; $u_k(t) = A_k(t)\cos(\phi_k(t))$ = the $k$th IMF; $\phi_k$ = the non-decreasing function; and $A_k$ = the

non-negative function. The constrained variational formulation can be modified as the following unconstrained pattern using an augmented Lagrangian method [78,79]:

$$L(\{u_k\}, \{\omega_k\}, \lambda) = \alpha \sum_{k=1}^{K} \left\| \partial t \left[ \left( \delta(t) + \frac{j}{\pi t} \right) * u_k(t) \right] e^{-j\omega_k t} \right\|_2^2 \\ + \left\| f(t) - \sum_{k=1}^{K} u_k(t) \right\|_2^2 + \left\langle \lambda(t), f(t) - \sum_{k=1}^{K} u_k(t) \right\rangle \tag{2}$$

where $L$ = the augmented Lagrangian; $\lambda$ = the Lagrange multiplier; and $\langle a, b \rangle$ = the scalar product of $a$ and $b$. Figure 3 explains the flowchart of the VMD algorithm [73].

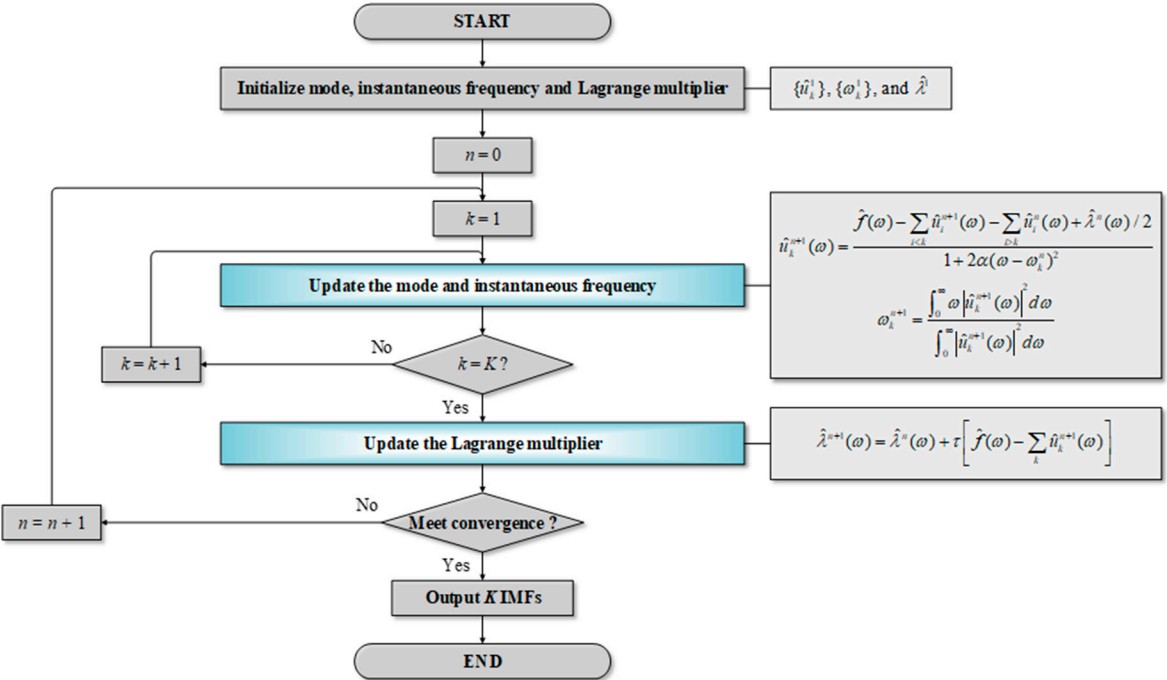

**Figure 3.** Flowchart for the variational mode decomposition (VMD) algorithm [73].

## 2.5. Hybrid Modeling Using DWT and VMD Approaches

DWT-based soft computing models (DWT-MLP and DWT-CCNN) are the hybrid models combined with the standalone models (MLP and CCNN) and DWT, respectively. In the same manner, VMD-based soft computing models (VMD-MLP and VMD-CCNN) conjugate the standalone models (MLP and CCNN) and VMD, respectively. Therefore, DWT- and VMD-based soft computing models consist of three steps (1). The training and testing dataset are decomposed into an approximation and multiple details using the DWT approach, and multiple IMFs using VMD, respectively (2). The standalone models (MLP and CCNN) are developed for each decomposed training dataset (3). The final predictions of DO concentration values are obtained by aggregating the sub-time series predicted from the standalone models (MLP and CCNN), respectively. Figure 4 represents the flowchart for DWT- and VMD-based soft computing modeling.

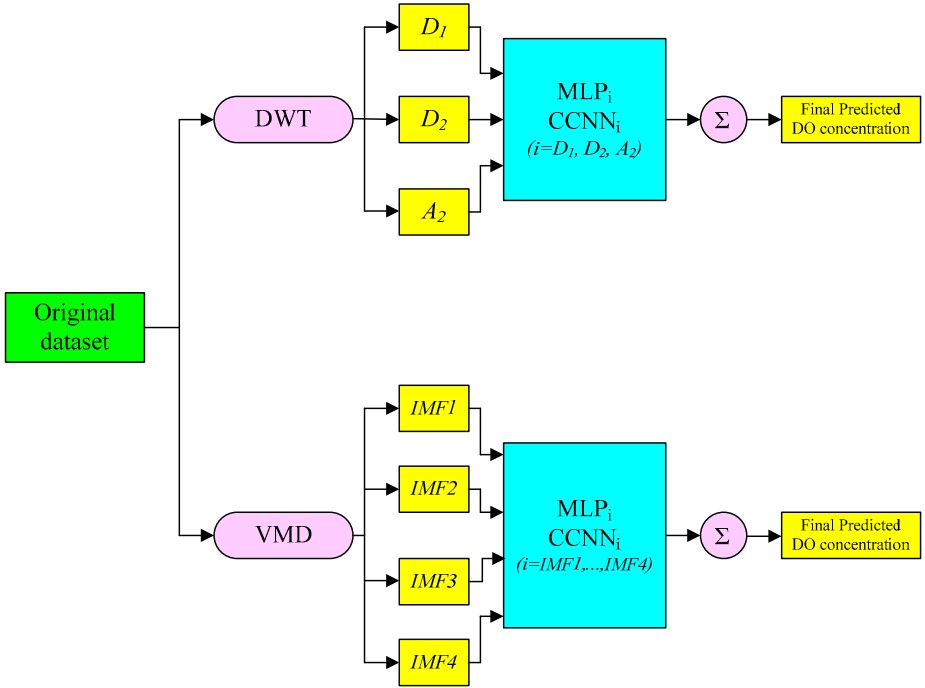

**Figure 4.** Flowchart for DWT- and VMD-based soft computing modeling.

### 2.6. Performance Evaluation of Models

Performance measures are assessed by comparing predicted values with their corresponding observed values using the following criteria:

I　　Root mean square error (RMSE):

$$RMSE = \sqrt{\frac{1}{n}\sum_{i=1}^{n}\left[DO_{obs} - DO_{pre}\right]^2}. \tag{3}$$

II　　Nash–Sutcliffe coefficient (NSE):

$$NSE = 1 - \frac{\sum_{i=1}^{n}\left[DO_{obs} - DO_{pre}\right]^2}{\sum_{i=1}^{n}\left[DO_{obs} - \overline{DO}_{pre}\right]^2}. \tag{4}$$

III　　Willmott's index of agreement (WI):

$$WI = 1 - \left[\frac{\sum_{i=1}^{n}\left(DO_{obs} - DO_{pre}\right)^2}{\sum_{i=1}^{n}\left(\left|DO_{pre} - \overline{DO}_{obs}\right| + \left|DO_{obs} - \overline{DO}_{obs}\right|\right)^2}\right]. \tag{5}$$

IV　　Mean absolute error (MAE):

$$MAE = \frac{1}{n}\sum_{i=1}^{n}\left|DO_{pre} - DO_{obs}\right|. \tag{6}$$

V    Coefficient of determination ($R^2$):

$$R^2 = \left( \frac{\left[ \sum_{i=1}^{n} DO_{obs}DO_{pre} - \frac{1}{n} \sum_{i=1}^{n} DO_{obs} \sum_{i=1}^{n} DO_{pre} \right]}{\left[ \sum_{i=1}^{n} DO_{obs}^2 - \frac{1}{n} \left[ \sum_{i=1}^{n} DO_{obs} \right]^2 \right] \left[ \sum_{i=1}^{n} DO_{pre}^2 - \frac{1}{n} \left[ \sum_{i=1}^{n} DO_{pre} \right]^2 \right]} \right), \tag{7}$$

where $DO_{obs}$ and $DO_{pre}$ are the observed and predicted values, respectively; $\overline{DO}_{obs}$ and $\overline{DO}_{pre}$ are the average of observed and predicted values, respectively; and $n$ is the length of time series data. The discrepancy between observed and predicted values can be shown using the RMSE criterion. A value of zero reflects perfect prediction. The RMSE criterion must be used for model evaluation to obtain accuracy in absolute units [80]. NSE is taken into account to evaluate the ability of predicting models [81]. If the squared difference between observed and predicted DO values is relatively large to concur with the variance in the observed DO values, then the NSE criterion will be zero. If the NSE criterion is negative, the results indicate that the observed mean is a better predictor than the model [81,82]. If the NSE criterion is equal to one, it indicates a perfect model [83]. The WI criterion varies between zero and one. WI calculates the ratio of mean square error (MSE) and can provide an advantage over the RMSE [84–86]. The MAE criterion can provide better information for a model's prediction. The MAE cannot be weighted towards higher or lower magnitudes. However, it evaluates all derivations from observed DO values in an equal manner [87].

## 3. Case Study

In this study, the fluctuations of the DO concentration independent of some parameters (e.g., chloride (Cl), nitrogen oxides (NOx), total dissolved solid (TDS), potential of hydrogen (pH), and water temperature (WT)) were selected as a case study. This study area was located in the southern east of Florida with a latitude of 28°32'33.864'' N and a longitude of 80°56'33.428'' W (Figure 5). All data numbered 232 records along about 12 years (1996–2013). The data were arbitrarily divided into two parts for training and testing phases. The training datasets were chosen at 80% ($n = 186$) of the data length and the testing datasets covered the remaining 20% ($n = 46$).

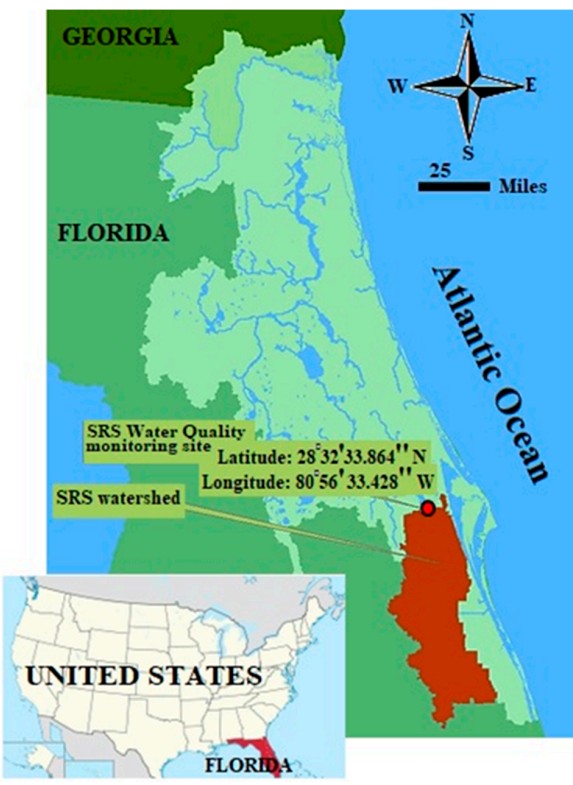

**Figure 5.** Location of the study area and the water quality monitoring station.

## 4. Application and Results

### 4.1. Setting up the Standalone Models

The statistical parameters of the collected dataset are presented in Table 1. A considerable wide domain of data values can be observed in Table 1 (e.g., max of NOx = 0.46 mg/L and max of TDS = 1950.0). This issue implies that before importing the data to the standalone models (MLP and CCNN), they should be standardized between specific ranges (e.g., from 0 to 1). As can be seen from Table 1, the pH was the highest correlated parameter to the DO concentration (correlation coefficient (CC) = 0.760). It was followed by NOx (CC = 0.554) and then by WT with a negative value of the correlation coefficient (CC = −0.544), which denoted the reverse effect of WT on the DO concentration. Based on the coefficient of variation, the WT data were least dispersed, and the NOx data were most sporadic. Several combinations for setting up the standalone models (MLP and CCNN) regarding the input vectors (e.g., consisting of five chemical characteristic parameters of chloride (Cl), nitrogen oxides (NOx), total dissolved solid (TDS), potential of hydrogen (pH), and water temperature (WT)) can be created. Bear in mind that the only value in the output layer always corresponded to the DO concentration.

**Table 1.** Summary of statistics parameters of input and output variables (*n* = 232).

|  | Variable | Unit | Min | Max | Median | Mean | SD | CV | CC |
|---|---|---|---|---|---|---|---|---|---|
|  | Cl | mg/L | 40.000 | 850.000 | 200.000 | 247.649 | 164.973 | 0.666 | 0.418 |
|  | NOx | mg/L | 0.001 | 0.460 | 0.072 | 0.089 | 0.087 | 0.978 | 0.554 |
| Input | TDS |  | 134.000 | 1950.000 | 559.500 | 642.362 | 355.163 | 0.553 | 0.394 |
|  | pH | Standard Units | 6.100 | 8.220 | 7.180 | 7.148 | 0.421 | 0.059 | 0.760 |
|  | WT | °C | 8.980 | 32.220 | 23.510 | 232.378 | 5.211 | 0.022 | −0.544 |
| Output | DO | mg/L | 0.090 | 11.480 | 5.920 | 5.485 | 2.565 | 0.468 | 1.000 |

Note: CV: the coefficient of variation, and CC: the coefficient correlation between inputs and DO.

In this study, six different input combinations for constructing the models, including five chemical characteristic parameters (combination ID = I) and even one quality parameter (combination ID = V and combination ID = VI), based on the highest positive and negative values of CC in Table 1 were constructed (see Table 2).

**Table 2.** Input combinations for structuring the MLP and CCNN models.

| ID of the Input Combination | Input | Input No. | Output |
|---|---|---|---|
| I | Cl, NOx, TDS, pH, WT | 5 | DO |
| II | Cl, NOx, pH, WT | 4 | DO |
| III | NOx, pH, WT | 3 | DO |
| IV | WT, NOx | 2 | DO |
| V | WT | 1 | DO |
| VI | PH | 1 | DO |

Constructing the ANN architecture involves the creation of the ANN topology and training parameters, such as the number of neurons in the hidden layer(s). Looking back at Table 2, the first step of ANN architecture for determining the input combinations was already completed. In the second step, the main structure of ANN in terms of the number of layers should be specified. Based on the reports for the capability of the one hidden layer supervised neural networks in simulating complex phenomena [88], this study adopted a one hidden layer architecture for the standalone models (e.g., MLP and CCNN). Finally, the optimal number of neurons in the hidden layer was determined using the MSE criterion by a trial and error approach (see the third column of Tables 3–5).

**Table 3.** Performance statistics of various input combination using MLP and CCNN models.

| Model | ID | Topology | Training Phase | | | | | Testing Phase | | | | |
|---|---|---|---|---|---|---|---|---|---|---|---|---|
| | | | RMSE (mg/L) | NSE | WI | $R^2$ | MAE (mg/L) | RMSE (mg/L) | NSE | WI | $R^2$ | MAE (mg/L) |
| MLP | I | 5-2-1 | 1.211 * | 0.778 * | 0.933 * | 0.781 * | 0.924 * | 1.425 | 0.663 | 0.897 | 0.769 | 1.126 |
| | II | 4-2-1 | 1.237 | 0.768 | 0.930 | 0.769 | 0.938 | 1.359 | 0.694 | 0.902 | 0.774 | 1.062 |
| | III ** | 3-4-1 | 1.306 | 0.742 | 0.918 | 0.752 | 0.994 | 1.261 * | 0.736* | 0.919 * | 0.801 * | 0.989 * |
| | IV | 2-4-1 | 1.830 | 0.493 | 0.817 | 0.496 | 1.437 | 1.427 | 0.662 | 0.868 | 0.707 | 1.072 |
| | V | 1-1-1 | 2.138 | 0.308 | 0.676 | 0.310 | 1.810 | 2.007 | 0.332 | 0.684 | 0.335 | 1.633 |
| | VI | 1-1-1 | 1.585 | 0.620 | 0.871 | 0.620 | 1.264 | 2.138 | 0.242 | 0.741 | 0.568 | 1.773 |
| CCNN | I | 5-1-1 | 1.227 | 0.772 | 0.933 | 0.773 | 0.953 | 1.172 | 0.772 | 0.935 | 0.795 | 0.790 |
| | II | 4-1-1 | 1.196 * | 0.783 * | 0.937 * | 0.785 * | 0.881 * | 0.581 | 0.773 | 0.935 | 0.801 | 0.195 |
| | III ** | 3-2-1 | 1.245 | 0.765 | 0.931 | 0.767 | 0.922 | 0.550 * | 0.797 * | 0.942 * | 0.825 * | 0.185 * |
| | IV | 2-3-1 | 1.760 | 0.531 | 0.837 | 0.531 | 1.341 | 0.674 | 0.695 | 0.896 | 0.705 | 0.246 |
| | V | 1-10-1 | 1.816 | 0.501 | 0.806 | 0.502 | 1.460 | 0.926 | 0.424 | 0.798 | 0.452 | 0.385 |
| | VI | 1-2-1 | 1.579 | 0.622 | 0.875 | 0.624 | 1.251 | 0.757 | 0.615 | 0.869 | 0.654 | 0.297 |

Note: * shows the best performance for each column; ** stands for introducing the best model.

**Table 4.** Performance statistics using DWT-MLP-III and DWT-CCNN-III models.

| Model | DWT | Topology | Training Phase | | | | | Testing Phase | | | | |
|---|---|---|---|---|---|---|---|---|---|---|---|---|
| | | | RMSE (mg/L) | NSE | WI | $R^2$ | MAE (mg/L) | RMSE (mg/L) | NSE | WI | $R^2$ | MAE (mg/L) |
| MLP-III | C6 | 9-27-1 | 0.523 | 0.958 | 0.989 | 0.958 | 0.345 | 0.364 | 0.978 | 0.994 | 0.978 | 0.291 |
| | C12 | 9-26-1 | 0.418 | 0.974 | 0.993 | 0.974 | 0.314 | 0.178 | 0.979 | 0.995 | 0.979 | 0.066 |
| | C18 | 9-21-1 | 0.441 | 0.971 | 0.992 | 0.971 | 0.343 | 0.202 | 0.973 | 0.993 | 0.975 | 0.072 |
| | D6 | 9-22-1 | 0.437 | 0.971 | 0.993 | 0.971 | 0.310 | 0.161 | 0.983 | 0.996 | 0.983 | 0.061 |
| | D12 | 9-22-1 | 0.346 | 0.982 | 0.995 | 0.982 | 0.255 | 0.180 | 0.978 | 0.995 | 0.979 | 0.066 |
| | D18 | 9-28-1 | 0.447 | 0.970 | 0.992 | 0.970 | 0.336 | 0.223 | 0.967 | 0.992 | 0.970 | 0.086 |
| | S6 | 9-22-1 | 0.437 | 0.971 | 0.993 | 0.971 | 0.310 | 0.161 | 0.983 | 0.996 | 0.983 | 0.061 |
| | S12 | 9-28-1 | 0.420 | 0.973 | 0.993 | 0.973 | 0.308 | 0.184 | 0.977 | 0.994 | 0.977 | 0.069 |
| | S18 | 9-37-1 | 0.434 | 0.971 | 0.993 | 0.972 | 0.294 | 0.186 | 0.977 | 0.994 | 0.977 | 0.068 |
| CCNN-III | C6 | 9-0-1 | 1.341 | 0.727 | 0.914 | 0.730 | 1.035 | 1.360 | 0.693 | 0.907 | 0.739 | 1.055 |
| | C12 | 9-0-1 | 1.335 | 0.730 | 0.916 | 0.735 | 1.041 | 0.656 | 0.711 | 0.913 | 0.747 | 0.250 |
| | C18 | 9-1-1 | 1.217 | 0.776 | 0.935 | 0.780 | 0.942 | 0.611 | 0.749 | 0.927 | 0.794 | 0.219 |
| | D6 | 9-2-1 | 1.216 | 0.776 | 0.934 | 0.778 | 0.956 | 0.605 | 0.754 | 0.928 | 0.792 | 0.219 |
| | D12 | 9-1-1 | 1.245 | 0.765 | 0.929 | 0.768 | 0.971 | 0.659 | 0.708 | 0.918 | 0.757 | 0.244 |
| | D18 | 9-0-1 | 1.330 | 0.732 | 0.917 | 0.735 | 1.028 | 0.869 | 0.709 | 0.912 | 0.755 | 0.245 |
| | S6 | 9-2-1 | 1.216 | 0.776 | 0.934 | 0.778 | 0.956 | 0.605 | 0.754 | 0.928 | 0.792 | 0.219 |
| | S12 | 9-0-1 | 1.336 | 0.730 | 0.915 | 0.734 | 1.046 | 0.683 | 0.687 | 0.903 | 0.728 | 0.258 |
| | S18 | 9-1-1 | 1.256 | 0.761 | 0.929 | 0.763 | 0.958 | 0.612 | 0.749 | 0.928 | 0.787 | 0.207 |

**Table 5.** Performance statistics using VMD-MLP-III and VMD-CCNN-III models.

| Model | VMD | Topology | Training Phase | | | | | Testing Phase | | | | |
|---|---|---|---|---|---|---|---|---|---|---|---|---|
| | | | RMSE (mg/L) | NSE | WI | $R^2$ | MAE (mg/L) | RMSE (mg/L) | NSE | WI | $R^2$ | MAE (mg/L) |
| MLP-III | $K = 3, \alpha = 5$ | 9-22-1 | 0.354 | 0.981 | 0.995 | 0.981 | 0.270 | 0.359 | 0.979 | 0.995 | 0.979 | 0.280 |
| | $K = 4, \alpha = 5$ | 12-25-1 | 0.335 | 0.983 | 0.996 | 0.983 | 0.240 | 0.146 | 0.986 | 0.996 | 0.986 | 0.054 |
| | $K = 4, \alpha = 10$ | 12-19-1 | 0.257 | 0.990 | 0.997 | 0.990 | 0.191 | 0.107 | 0.992 | 0.998 | 0.993 | 0.034 |
| CCNN-III | $K = 3, \alpha = 5$ | 9-0-1 | 1.312 | 0.739 | 0.920 | 0.744 | 1.003 | 1.307 | 0.717 | 0.916 | 0.762 | 0.965 |
| | $K = 4, \alpha = 5$ | 12-0-1 | 1.322 | 0.735 | 0.921 | 0.737 | 1.026 | 0.611 | 0.750 | 0.926 | 0.792 | 0.228 |
| | $K = 4, \alpha = 10$ | 12-0-1 | 1.327 | 0.733 | 0.921 | 0.738 | 1.036 | 0.597 | 0.761 | 0.929 | 0.791 | 0.218 |

## 4.2. Performance of Standalone Models

A statistical summary of the DO concentration performance using the standalone models (MLP and CCNN) is given in Table 3. Results of the MLP model with higher (NSE, WI, and $R^2$) and lower (RMSE and MAE) values indicated that the first (MLP-I) and third (MLP-III) input combinations acted better than the others for the training and testing phases, respectively. Since, however, choosing the best model is always based on the performance of the testing phase, the MLP model with the third combination topology (MLP-III) was selected as the best model. Similar interpretation can be done for the CCNN models. While the second combination (CCNN-II) provided the best performance for the training phase, the third input combination (CCNN-III) gave the best results for the testing phase. All the statistics were enriched in the CCNN-III model in comparison to the MLP-III model. A general comparison of the MLP-III and CCNN-III models revealed that the CCNN-III model yielded the better predictions than the MLP-III model. The RMSE, NSE, WI, $R^2$, and MAE criteria for the testing phase of the CCNN-III model were improved by 129%, 8%, 2%, 3%, and 434% compared with the MLP-III model, respectively.

The variations of the observed versus predicted DO concentration values are depicted in Figure 6a,b. Visual analysis confirmed the better results of the CCNN-III model compared to those of the MLP-III model with he observed values. Figure 7a,b provides scatter plots of the testing dataset between the observed and predicted values using the MLP-III and CCNN-III models. From visual interpretation of Figure 7a,b, it can be understood that the dots in the MLP-III plot were a bit more sporadic than those in the CCNN-III plot. In addition, based on the coefficient of determination and the slope of the trend lines to the unity, it can be concluded that the CCNN-III model acted better in predicting the DO concentration.

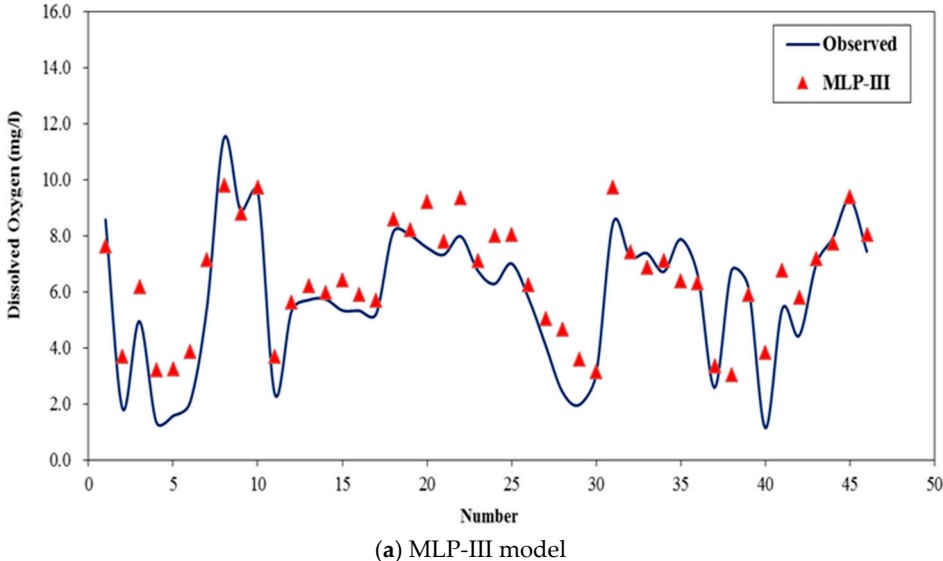

(**a**) MLP-III model

**Figure 6.** *Cont*.

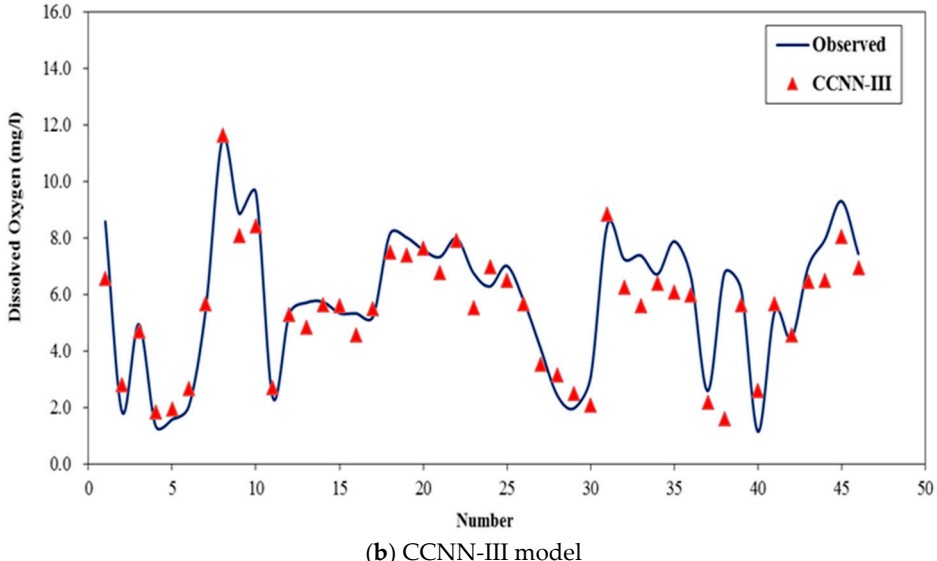

(**b**) CCNN-III model

**Figure 6.** Comparison of observed and predicted DO values using standalone models for the testing phase.

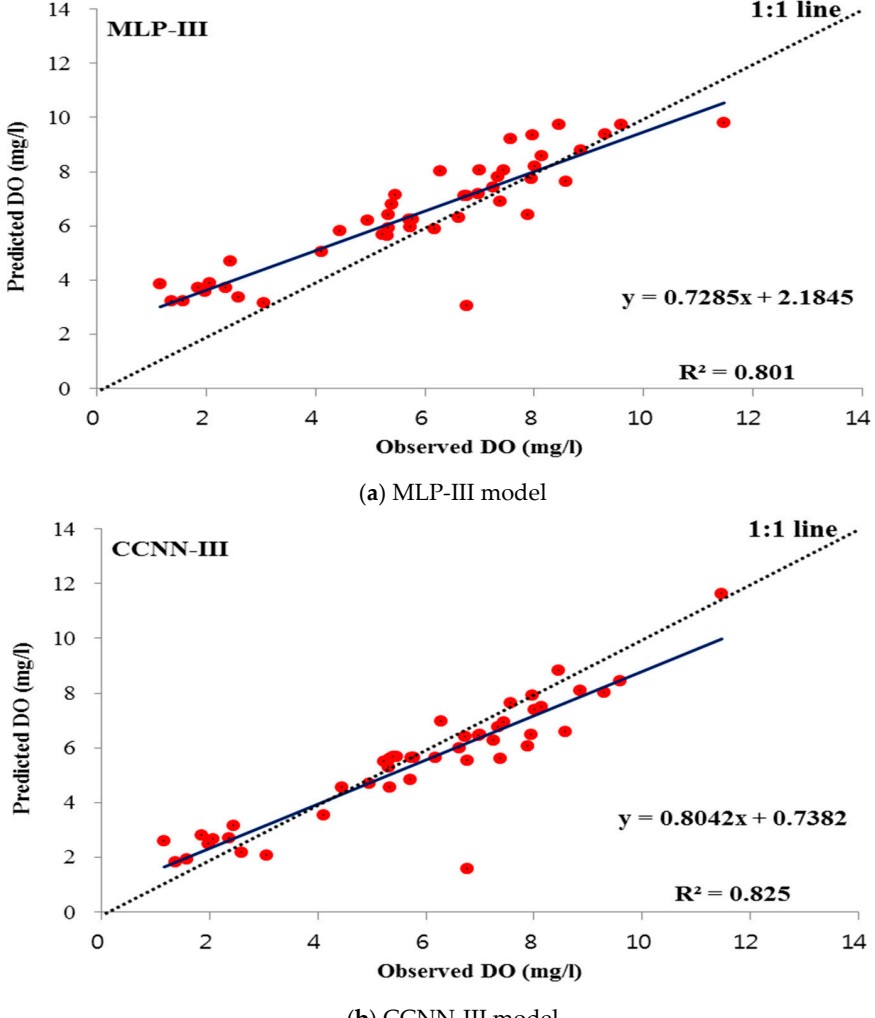

(**a**) MLP-III model

(**b**) CCNN-III model

**Figure 7.** Scatter plots of observed and predicted DO values using standalone models for the testing phase.

Figure 8 displays the error residual in mg/L over the testing phases of the MLP-III and CCNN-III modeling. It explained that the residuals of local peaks were relatively insignificant for the CCNN-III model, and tended to overestimate the DO concentration values; whereas the MLP-III model had larger residuals, and tended to underestimate the DO concentration values.

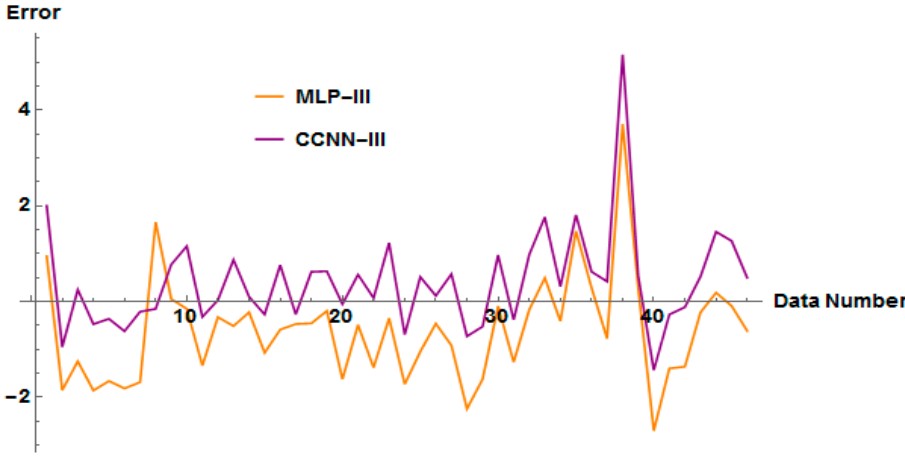

**Figure 8.** Error residual values for the standalone models for the testing phase.

### 4.3. Performance of Hybrid Models

### 4.3.1. DWT-Based Soft Computing Models

To decompose the input dataset using the DWT algorithm, the optimal level of decomposition (*L*) should be selected. In this study, Equation (8) was used to calculate the optimal level of decomposition [68,70,72]. Although the optimal level of decomposition can be implemented using a trial-and-error method, it is time-consuming and a waste of energy:

$$L = \text{int}[\log(N)], \tag{8}$$

where $N$ is the length of the time series, $\text{int}[k]$ returns the integer portion of $k$, and $k$ is a real number.

In this study, $L = 2$ was determined using Equation (8). In addition, mother wavelets have to be set before DWT-based soft computing models are employed. Using different mother wavelets, the dataset was decomposed with a details ($D_1$ and $D_2$) and an approximation ($A_2$) components for individual input data [72,75]. For the DWT algorithm, Daubechies, Symmlets, and Coiflets have been frequently used as mother wavelets in previous studies [72,75,89,90]. Therefore, the underlying mother wavelets, including Coiflet-6 (C6), Coiflet-12 (C12), Coiflet-18 (C18), Daubechies-6 (D6), Daubechies-12 (D12), Daubechies-18 (D18), Symmlet-6 (S6), Symmlet-12 (S12), and Symmlet-18 (S18), were implemented. For each DWT-based soft computing model, the optimal mother wavelet yielding the best model performance was recommended. Figure 9 shows an approximation and details decomposed using the Symmlet-6 (S6) mother wavelet for the original water temperature (WT).

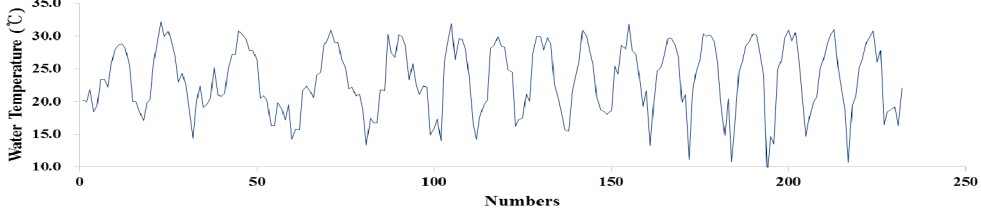

(**a**) Original water temperature

**Figure 9.** *Cont.*

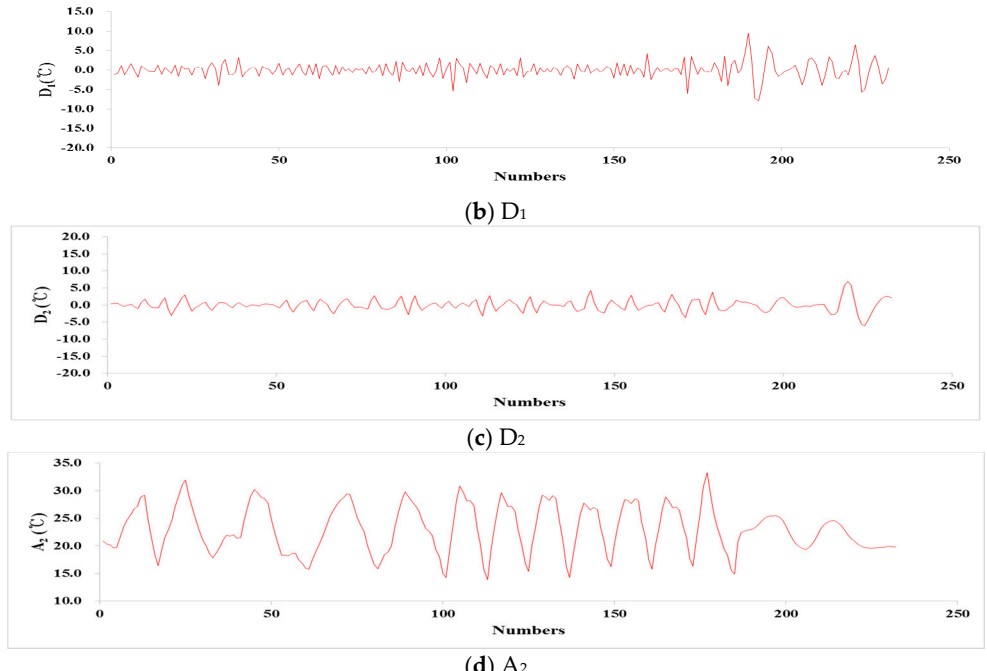

**Figure 9.** Original water temperature and sub series (details ($D_1$ and $D_2$) and approximation ($A_2$) components) decomposed using Symmlet-6 (S6) mother wavelet.

Table 4 gives the performance statistics using the DWT-MLP-III and DWT-CCNN-III models during the training and testing phases. Tables 3 and 4 suggested that all of the DWT-MLP-III models improved the performance of the MLP-III model significantly, while all of the DWT-CCNN-III models did not improve the performance of the CCNN-III model during the testing phase. In addition, the DWT-MLP-III (D6) and DWT-MLP-III (S6) (e.g., RMSE = 0.161 (mg/L), NSE = 0.983, WI = 0.996, $R^2$ = 0.983, and MAE = 0.061 (mg/L) for D6 and S6) models produced the best results among all of the DWT-MLP-III models during the testing phase. The combination of the DWT into the MLP-III model could confirm the model accuracy for the prediction of the DO concentration. However, the CCNN-III model provided better results than any of the DWT-CCNN-III models. A comparison explained that all the DWT-MLP-III models yielded better results compared with all the DWT-CCNN-III models.

Figure 10a,b shows the scatter plots of the testing dataset between the observed and predicted values using the DWT-MLP-III (S6) and DWT-CCNN-III (S6) models. From visual interpretation of Figure 10a,b, it can be explained that the dots in the DWT-CCNN-III (S6) plot were extremely sporadic compared to those in the DWT-MLP-III (S6) plot. In addition, based on the coefficient of determination and the slope of trend lines to the unity, it can be concluded that the DWT-MLP-III (S6) model acted better in predicting the DO concentration.

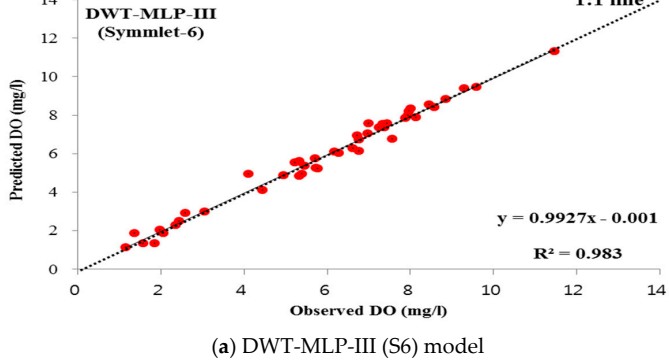

(**a**) DWT-MLP-III (S6) model

**Figure 10.** *Cont.*

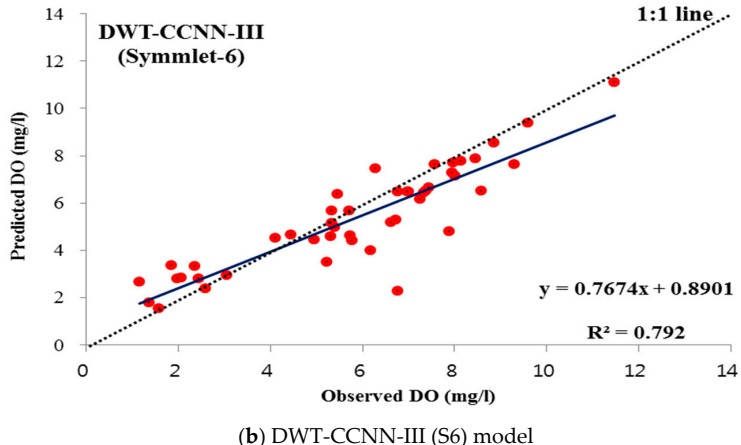

(**b**) DWT-CCNN-III (S6) model

**Figure 10.** Scatter plots of observed and predicted DO values using DWT-based soft computing models for the testing phase.

### 4.3.2. VMD-Based Soft Computing Models

To decompose the input dataset using the VMD algorithm, the number of IMFs ($K$) and the quadratic penalty factor ($\alpha$) have to be implemented in advance. In this study, different sets of parameters were investigated, and three sets of parameters with higher correlations between the original and predicted data (i.e., the aggregation of decomposed series) were selected. The three sets were ($K$, $\alpha$) = {(3, 5), (4, 5), (4, 10)}. Among them, the optimal parameters ($K = 4$ and $\alpha = 10$) yielding the best performance of the VMD-based soft computing models were chosen finally. Figure 11 shows the original WT series and the IMFs decomposed using the VMD algorithm ($K = 4$ and $\alpha = 10$).

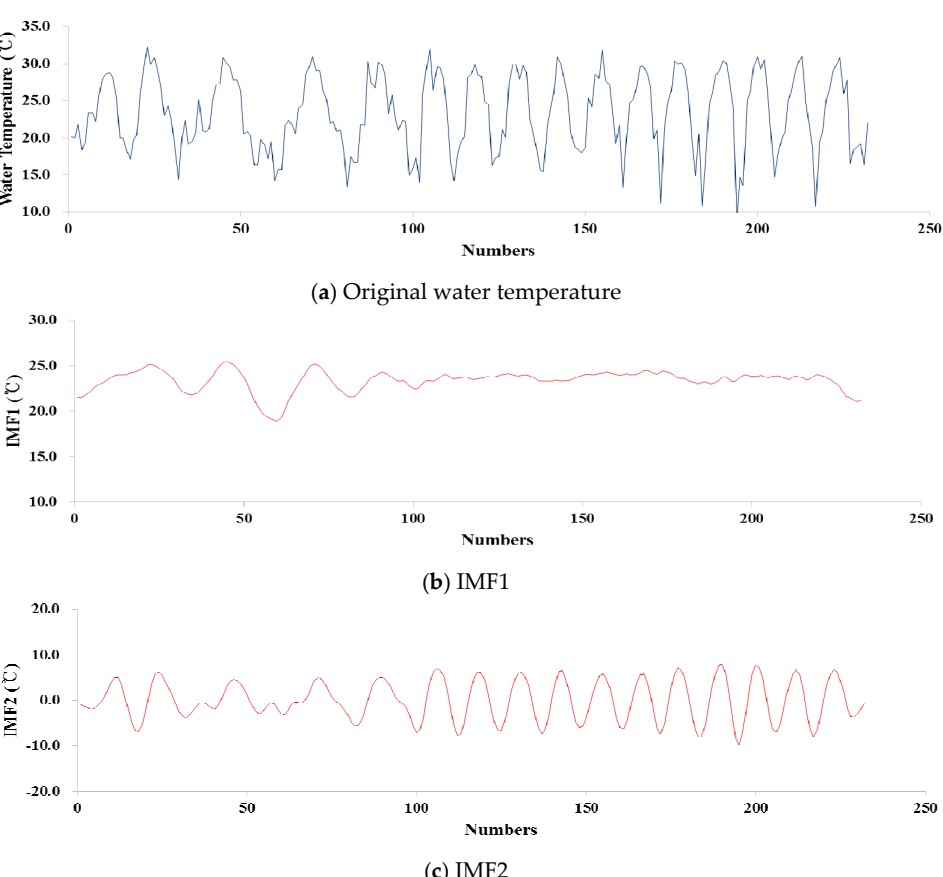

(**a**) Original water temperature

(**b**) IMF1

(**c**) IMF2

**Figure 11.** *Cont*.

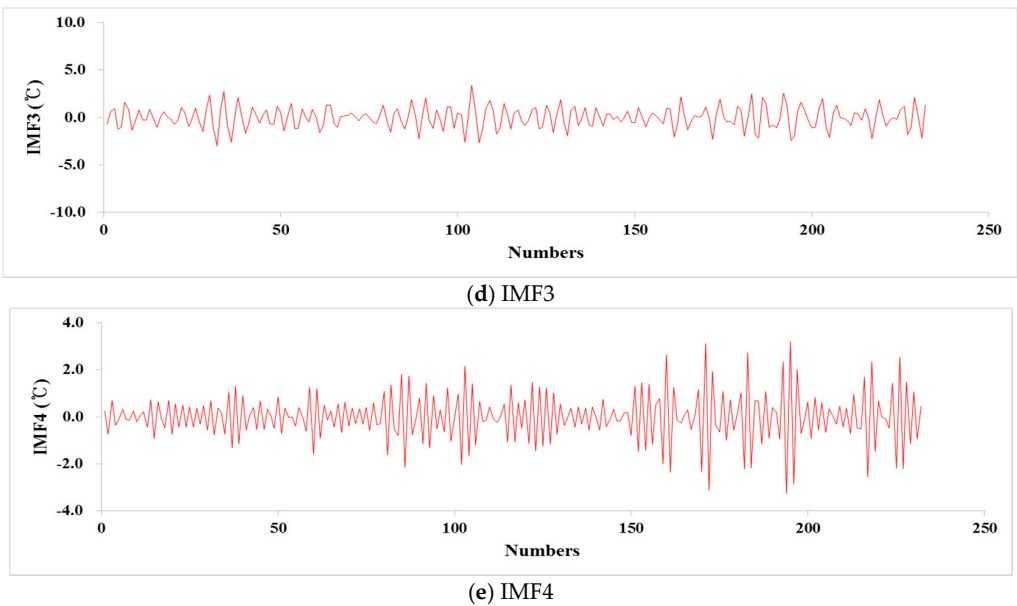

**Figure 11.** Original water temperature and intrinsic mode functions (IMF1, IMF2, IMF3, and IMF4) decomposed using the VMD algorithm ($K = 4$ and $\alpha = 10$).

Table 5 shows the performance statistics using the VMD-MLP-III and VMD-CCNN-III models during the training and testing phases. Tables 3 and 5 indicate that all of the VMD-MLP-III models enhanced the performance of the MLP-III model enormously, while all of the VMD-CCNN-III models did not enhance the performance of the CCNN-III model during the testing phase. In addition, the VMD-MLP-III ($K = 4$ and $\alpha = 10$) model (e.g., RMSE = 0.107 (mg/L), NSE = 0.992, WI = 0.998, $R^2$ = 0.993, and MAE = 0.034 (mg/L)) provided the best results among all of the VMD-MLP-III models during the testing phase. The conjugation of VMD into the MLP-III model ensured the model accuracy for predicting the DO concentration. However, the CCNN-III model provided better results compared to all of the VMD-CCNN-III models. A comparison explained that all of the VMD-MLP-III produced better results compared with all of the VMD-CCNN-III models.

Tables 4 and 5 explain that the statistical results of the DWT- and VMD-MLP-III models showed similar statistical patterns. A comparison of the best models revealed, however, that the VMD-MLP-III ($K = 4$ and $\alpha = 10$) model yielded slightly better results than the DWT-MLP-III (D6 and S6) model. Figure 12a,b shows scatter plots of the testing dataset between the observed and predicted values using the VMD-MLP-III ($K = 4$ and $\alpha = 10$) and VMD-CCNN-III ($K = 4$ and $\alpha = 10$) models. From visual interpretation of Figure 12a,b, it can be explained that the dots in the VMD-CCNN-III ($K = 4$ and $\alpha = 10$) plot were extremely sporadic compared to those in the VMD-MLP-III ($K = 4$ and $\alpha = 10$) plot. Based on the coefficient of determination and the slope of trend lines to the unity, it can be concluded that the VMD-MLP-III ($K = 4$ and $\alpha = 10$) model acted better in predicting the DO concentration.

In general, various studies have reported that the combination of soft computing models and decomposition approaches improved the accuracy and reliability of model performance for predicting DO concentration [12,47–52]. Even if the CCNN model showed the outstanding performance for standalone models, the combination of the CCNN model and decomposition approaches cannot improve the model performance. The special model structure (e.g., adding hidden nodes) can prevent the model performance of complex nonlinear signals. To confirm the model performance, continued studies are required using different data, soft computing models, and decomposition approaches for predicting diverse environmental parameters (e.g., BOD, COD, TP, and TN etc.).

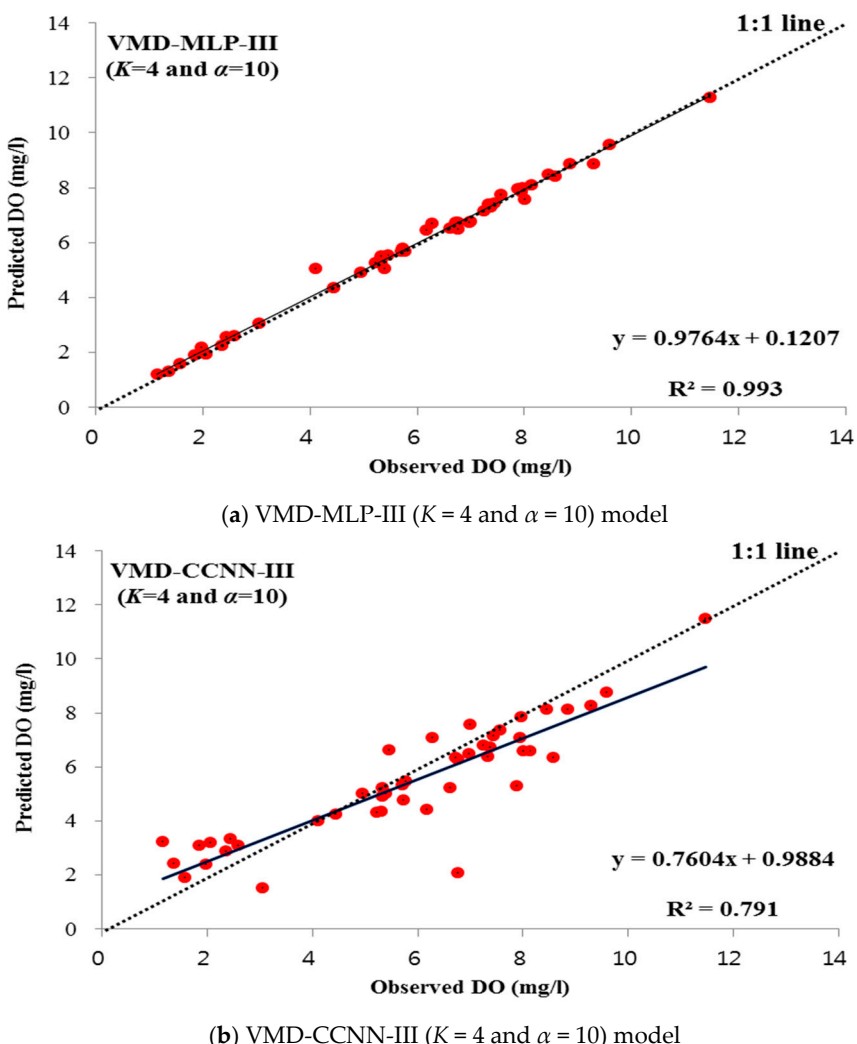

(**a**) VMD-MLP-III ($K$ = 4 and $\alpha$ = 10) model

(**b**) VMD-CCNN-III ($K$ = 4 and $\alpha$ = 10) model

**Figure 12.** Scatter plots of observed and predicted DO values using VMD-based soft computing models for the testing phase.

### 4.4. Diagnostic Analysis

In this study, three diagnostic analysis methods (i.e., Taylor diagram and violin plot) were considered for visual evaluation of the model performance.

#### 4.4.1. Taylor Diagram

A polar plot presented by Taylor [91] was drawn for obtaining a visual understanding of model performance. It has the ability to highlight the goodness of model performance in comparison to observed values. The Taylor diagram depicts three statistics: (1) Correlation coefficient (the azimuth angle), (2) normalized standard deviation (radial distance from the origin), and (3) RMSE (distance from the reference observed point). A perfect matching of the predicted results is identified as a complete overlay by the reference point with the correlation coefficient equal to unity and the exact amplitude of variations compared with observations [91–94].

Figure 13a shows the Taylor diagram of the standalone models (MLP and CCNN). In the case of the best models, the diagram shows that the CCNN-III model had a lower RMSE than the MLP-III model. Although the correlation coefficients and standard deviations of the predicted data for both models were less than the observations, the node representing the CCNN-III model was closer to the observation node. Figure 13b shows the Taylor diagram of the hybrid models. The diagram shows

that the DWT-MLP-III (S6) and VMD-MLP-III ($K = 4$ and $\alpha = 10$) models had a lower RMSE than the DWT-CCNN-III (S6) and VMD-CCNN-III ($K = 4$ and $\alpha = 10$) models. Although the correlation coefficients and standard deviations of the predicted data of the DWT-MLP-III (S6) and VMD-MLP-III ($K = 4$ and $\alpha = 10$) models were less than observations, the node representing the VMD-MLP-III ($K = 4$ and $\alpha = 10$) model was closer to the observation node.

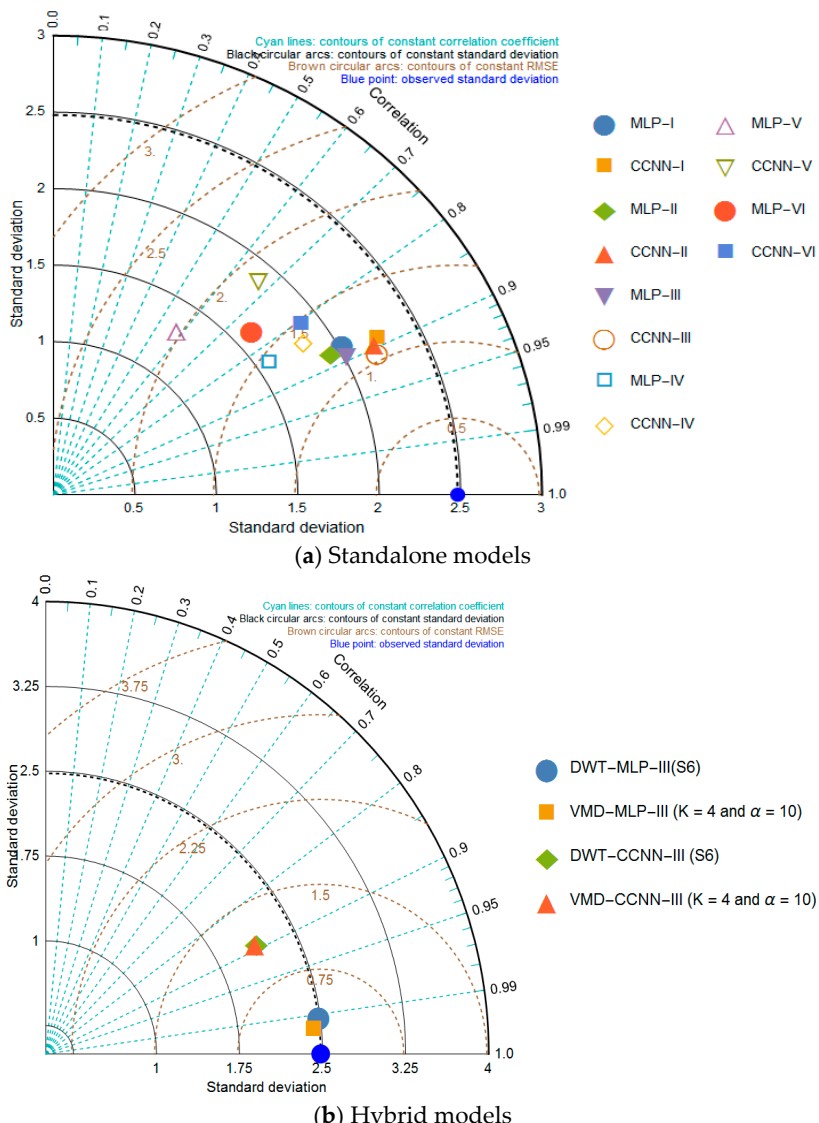

(**a**) Standalone models

(**b**) Hybrid models

**Figure 13.** Taylor diagrams for evaluating the performance of the applied models.

### 4.4.2. Violin Plot

As a further diagnostic tool, the violin plot is utilized in Figure 14a,b to assess the predicted results of the developed models for DO concentration. The violin plot, which has the ability to indicate the probability distribution of an observed and predicted dataset, is categorized as a box plot with the integration of the kernel density plot [95]. Based on the legends of Figure 14a, the median of the observed data was predicted by the MLP-III accurately (6.173 vs. 6.403), while the 25th and 75th percentiles in the CCNN-III had a better fit than the MLP-III. In addition, the MLP-III model overestimated the minimum, 25th percentile, median, and 75th percentile range of the DO concentration, whereas the CCNN-III model underestimated the 25th percentile, median, and 75th percentile range of the DO concentration. Overall, the violin plots indicated that the CCNN-III model performed better than the MLP-III model. Figure 14b explains that the VMD-MLP-III ($K = 4$ and $\alpha = 10$) model

overestimated the minimum, 25th, and median of the DO concentration, while the DWT-MLP-III (S6) underestimated the minimum, median, 75th percentile, and maximum of the DO concentration. Overall, the violin plots indicated that the VMD-MLP-III ($K = 4$ and $\alpha = 10$) model performed slightly better than the DWT-MLP-III (S6) model.

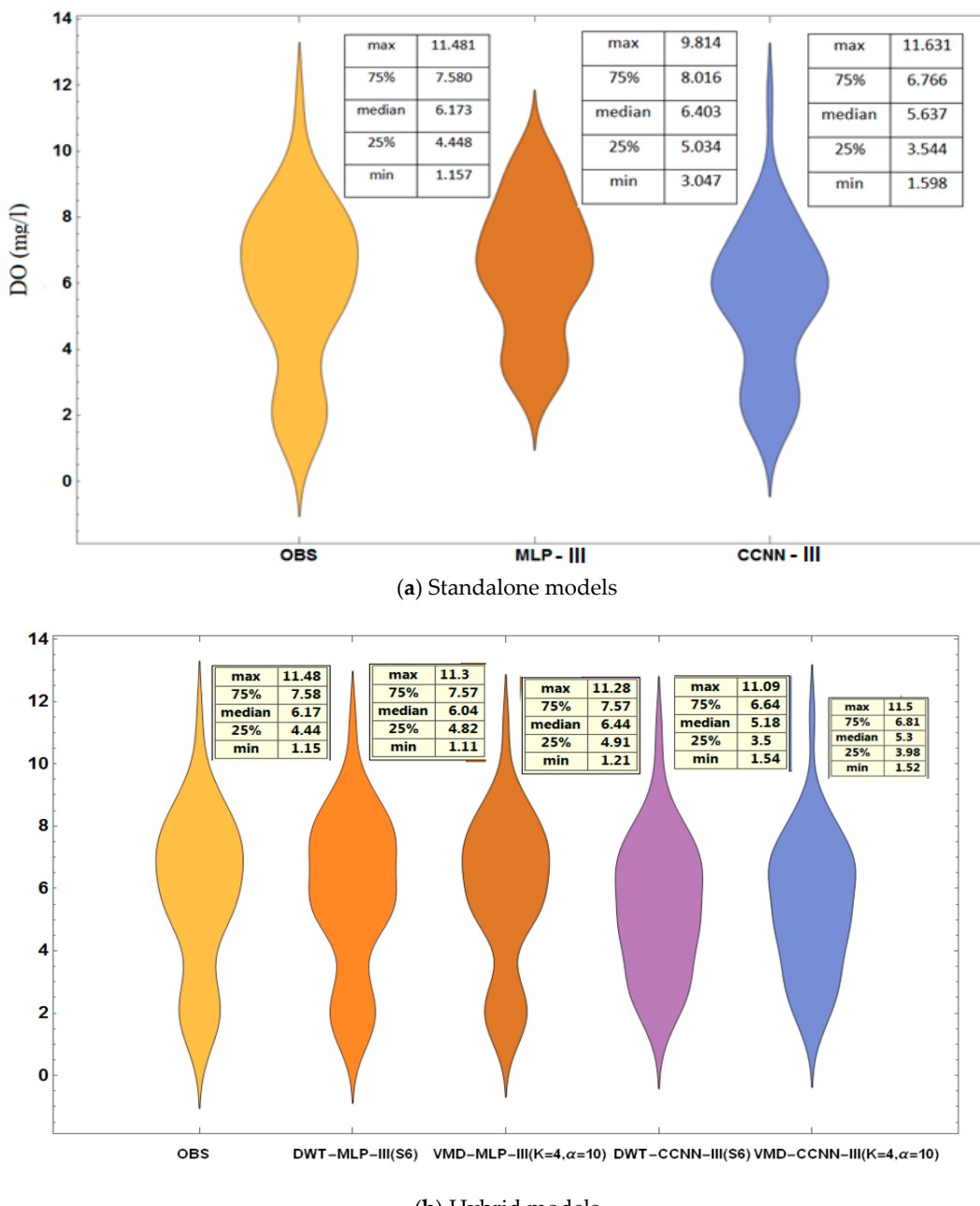

(**a**) Standalone models

(**b**) Hybrid models

**Figure 14.** Violin plots of observed and predicted DO concentration.

## 5. Conclusions

This study investigated the accuracy of two heuristic (MLP and CCNN) and decomposition (DWT and VMD) approaches for predicting dissolved oxygen (DO) concentration. To achieve this goal, the DO concentration and five chemical input parameters (Cl, NOx, TDS, pH, and WT) in the St. Johns River, Florida, USA, were used. For training and testing the developed models, the total dataset was divided into 80% and 20%, respectively. Several statistical indices (e.g., RMSE, NSE, WI,

$R^2$, and MAE) and diagnostic analyses (e.g., Taylor diagram and violin plot) were used to compare the developed models.

In the first stage, it was found that the CCNN-III (RMSE = 0.550 mg/L, NSE = 0.797, WI = 0.942, $R^2$ = 0.825, and MAE = 0.185 mg/L) and MLP-III (RMSE = 1.261 mg/L, NSE = 0.736, WI = 0.919, $R^2$ = 0.801, and MAE = 0.989 mg/L) provided the best results based on the standalone model category. In addition, a comparison suggested that the CCNN-III model performed better than the MLP-III model. In the second stage, however, it was found that the DWT-MLP-III (D6 and S6) (RMSE = 0.161 mg/L, NSE = 0.983, WI = 0.996, $R^2$ = 0.983, and MAE = 0.061 mg/L) and VMD-MLP-III ($K$ = 4 and $\alpha$ = 10) (RMSE = 0.107 mg/L, NSE = 0.992, WI = 0.998, $R^2$ = 0.993, and MAE = 0.034 mg/L) produced the best results based on the hybrid model category. Unfortunately, the CCNN-III did not improve the performance using the DWT and VMD approaches.

**Author Contributions:** M.Z.-K. conceived and designed the research; Y.S. implemented DWT and VMD; S.K. applied the CCNN model and wrote the original manuscripts; M.A.G. implemented the MLP model and performed analyses; S.S. and S.N. collected and analyzed the research data; N.W.K. analyzed the results; V.P.S. reviewed and edited the manuscripts.

**Funding:** A grant from a Strategic Research Project (20190153-001) funded by the Korea Institute of Civil Engineering and Building Technology.

**Acknowledgments:** The authors would like to appreciate that this research was supported by a grant from a Strategic Research Project (20190153-001) funded by the Korea Institute of Civil Engineering and Building Technology.

**Conflicts of Interest:** The authors declare no conflict of interest.

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
