# Peer review of "Can Decomposition Approaches Always Enhance Soft Computing Models? Predicting the Dissolved Oxygen Concentration in the St. Johns River, Florida"

_applsci, doi:10.3390/app9122534_

Reviewer 1 Report

The work presents an exhaustive comparative study of models to predict the concentration of dissolved oxygen. The various models are duly presented and described. The case study is very interesting and very well presented. The comparison of the models is exhaustive and with drawings. The analyzes and conclusions are pertinent and are supported by the statistical analysis performed.

It is only suggested that some graphics and images be improved, especially the legends, by standardizing fonts and sizes.

Reviewer 2 Report

This is an interesting study on predicting the dissolved oxygen concentration in the St. Johns River in Florida by using hybrid and standalone models. This study is well organized and authors have made a lot of efforts. I would recommend the manuscript be accepted for publication after lots of clarifications and improvements. Specific comments are as follows:

       Major comments:

1.     Figure 1, the circle symbols seem to have nothing to do with input layers, hidden layers and output layers. Where are the three layers in Fig. 1? What do Wji Wkj, B1, B2 mean in Figure 1a? They need to be clarified.

2.     For your designed experiments in Table 3, the Topology column, MLP-I is 5-2-1, however, CCNN-I is 5-1-1? Why the second number is different in the same ID number? For CCNN-V, the topology is 1-10-1. The second number is so high and why is that? The same comment as Table 4 and 5.

3.     Section 4.4.2 and Section 4.4.3, the two sections describe the same thing. You can keep one of them. Another question is in Fig. 14. In both subplots of Fig. 14, why ‘OBS’ showed negative values below zero? In Fig. 14(a), the maximum of ‘OBS’ is 11.481, why your plot exceeded 12 on y axis? Y axis mismatched in Fig. 14 (a) and (b). They need to be corrected and clarified. 

4.     DO data was obtained from water surface or bottom? How many data do you have? Is it a long-term data lasting several months every year from 1996 to 2013? Do you have any time series DO plot rather than the number vs DO concentration?

Minor comments

1.  In Introduction, the common water quality issues in the estuaries, for example, Caloosahatchee River Estuary (Xia et al., 2010), Florida and Perdido Bay (Xia and Jiang, 2015), and also your study area St. Johns River need to be added, especially the DO conditions.

Reference:

Xia, M., Craig, P.M., Schaeffer, B., Stoddard, A., Liu, Z., Peng, M., Zhang, H., Wallen, C.M., Bailey, N., Mandrup-Poulsen, J. (2010). Influence of physical forcing on bottom-water dissolved oxygen within Caloosahatchee River Estuary, Florida. Journal of Environmental Engineering, 136(10), 1032-1044.

Xia, M., & Jiang, L.* (2015). Influence of wind and river discharge on the hypoxia in a shallow bay. Ocean Dynamics, 65(5), 665-678.

      2. Minor writing errors need to be fixed. For example, Line 31, ‘R2’ should be ‘R2’.

Author Response

Please see the attachments

Round  2

Reviewer 2 Report

Thank you for giving me this opportunity to review this manuscript. This is a careful study on predicting the dissolved oxygen concentration in the St. Johns River in Florida by using hybrid and standalone models. Authors have made a lot of efforts and fixed all my comments. I would recommend the manuscript be accepted for publication.